# Predicting WEEE Generation Rates in Jordan Using Population Balance Model

**Feras Y. Fraige** [1,*] **, Laila A. Al-Khatib** [2] **and Mou'ath A. Al-Shaweesh** [2]

1 Mining & Mineral Engineering Department, Faculty of Engineering, Al-Hussein Bin Talal University, P.O. Box 25, Ma'an 71111, Jordan
2 Environmental Engineering Department, Faculty of Engineering, Al-Hussein Bin Talal University, P.O. Box 25, Ma'an 71111, Jordan
* Correspondence: ferasfraige@gmail.com or dr.feras@ahu.edu.jo; Tel.: +962-788564979 or +962-32179000 (ext. 8505); Fax: +962-32179050

**Abstract:** Waste generated from electric and electronic equipment (WEEE) is increasing rapidly due to the high demand for appliances, rapid product obsolescence, coupled with rapid economic growth, urbanization and technology advancement. Setting up a proper WEEE management system, which ensures better collection, treatment, recycling and control of transboundary movement of waste is crucial to increasing resource efficiency, improving sustainable production, use and consumption, and promoting the circular economy in Jordan. However, this system requires proper assessment of WEEE generation rates and reliable figures. Estimation of historical and future electric and electronic equipment put on market (EEE POM) and WEEE generation rates in Jordan have been achieved using the population balance model (PBM), logistic growth model (LGM) and Weibull distribution from 2000 to 2050. It is expected that the total disposal of appliances will reach about 1.6 million units (53 kt) in 2022, double this figure by 2044 and hit around 5 million units (175 kt) in 2050, with increasing WEEE generation rates. This is combined with the changing composition of WEEE with time. Thus, a rapid increase of WEEE in the near future is expected; this increase requires close monitoring and immediate response to tackle this hazardous waste.

**Keywords:** WEEE (E-waste); sustainability; modeling; population balance model; logistic growth model; Weibull distribution; Jordan; environment

## 1. Introduction

The production of electric and electronic equipment (EEE) is increasing rapidly at the global level [1–3]. This is driven by the high global consumption of EEE that increases by 2.5 million tons (Mt) annually [2]. Besides higher economic growth, urbanization, industry 4 and technology advancement have led to rapid product obsolescence over the last two decades [1]. Consequently, waste electric and electronic equipment (WEEE) is hitting a global record of 53.6 Mt in 2019, an increase of 21% compared with 2014 [2]. It is expected to double 2014 figures by 2030 [3].

Generation rates of WEEE vary considerably across the globe. In their report on global WEEE monitoring for 2020, Forti et al. [2] reported that Asia was in the lead in terms of WEEE generation figures of about 24.9 Mt, followed by the Americas (13.1 Mt), Europe (12 Mt), Africa (2.9 Mt) and Oceania (0.7 Mt). Based on the kilograms per inhabitant (kg/inh) indicator, Europe generated the most (16.2 kg/inh), followed by Oceania (16.1 kg/inh), the Americas (13.3 kg/inh), Asia (5.6 kg/inh) and lastly Africa (2.5 kg/inh). However, statistics are not even across different regions and countries due to multiple technical, socioeconomic and cultural factors as well as the existence of waste management systems and degree of implementation [2].

WEEE is a unique type of waste due to its diverse and complex composition. It contains valuable metals and hazardous materials [4]. From circularity perspectives, it

is regarded as a resource not a waste. Collected End of life (EoL) appliances should be returned to the circular path channels using circular flow strategies for recovery of valuables and materials with added value [5]. This is of paramount importance in mitigating the environmental effect of WEEE, increasing resource efficiency and fulfilling partially the eager of EEE industry for raw materials.

Although, estimated WEEE generation figures are increasing across the globe, collection and recycling levels are far from total demand of materials like low-volume metals and rare earth elements [2,6,7]. For example, significant quantities of EEE, such as mobile phones, are not collected for recycling or reuse across the EU [8]. On the other hand, evidence of encouraging alteration has been noticed in some industries; however, the transition to circularity has generally been slow [9–11]. This should be considered in setting/updating the policies and strategies dealing with WEEE.

There are many policies, legislations, regulations and initiatives that aim to manage WEEE in a sustainable manner. The Basel Convention forms a legislative framework for hazardous waste and controls its transboundary movements that are applicable to WEEE [12]. It aims at reducing the hazardous waste generation and promoting sustainable handling techniques at the place of origin. The EU WEEE Directive aims at contributing to sustainable production and consumption by setting prevention of WEEE creation as a priority [13]. It enhances the efficient use of resources and material recovery forms (such as reuse and recycling). Moreover, it endeavors to improve the environmental performance at the different levels of EEE life cycle. China—the largest producer and recipient of WEEE—approved the RoHS Directive in 2006 to control pollution of electronic products. It restricts the use of hazardous materials, such as mercury, in electronics [14]. It adopts effective policies, such as shared financial responsibility among customers, manufacturers and the government, and sets a legislative and institutional framework that includes a WEEE channel that consists of both informal sector (responsible of handling collection and reuse) and formal sector (responsible for dismantling and recycling). This energetic combination between formal and informal sectors led China to become a model for developing countries [15–17]. The situation in USA is quite different. Although USA is one of the largest WEEE producers, landfilling of WEEE and recycling without strict environmental regulations were allowed in the past. In 2011, the National Strategy for Electronics Stewardship (NSES) was applied to improve the design of electronic equipment and to enhance the management of WEEE [18].

On the other hand, numerous initiatives to tackle WEEE problems are active in the field, such as "solving the e-waste problem" [19], "Global e-Sustainability" [20], 3R initiative [21] and 6R initiatives [22]. Although all these policies, legislations and initiatives are aiming at an efficient and effective WEEE management system, EU WEEE Directive is still considered the focal reference for the regulations, policies concerned and global approved [5].

Some countries in the Middle East and North Africa (MENA) region have well-developed legal and regulatory frameworks in the field of solid and hazardous waste management, which can also be applied to WEEE [7]. Jordan recently approved WEEE instruction [23], while UAE applied principles of extended producer responsibility (EPR) legislation on WEEE [7]. However, tackling the WEEE issue in Jordan is still in the premature level, and areas of development are open wide. Reliable generation figures, efficient management systems, raising awareness, sustainable production and consumption, timely collection and the application of 3R initiative (reduce, reuse, recycle) in the circular supply chain design of EEE are of inevitable importance.

Optimized planning of WEEE policies, proper management of take-back systems and close monitoring of legislative implementation necessitate accurate assessment of WEEE [24]. So, a clear methodology should be framed to estimate national WEEE quantities to help implementing these targets.

Different techniques are used to quantify WEEE generation, including disposal analysis, projections analysis, factor models and input–output analysis [25–29]. Disposal analysis employs data about WEEE obtained from collection sites, treatment and recycling plants. The estimation of overall generation figures may require some assumptions and empirical

data for different disposal paths. While projection models predict the future development of WEEE generation using extrapolation techniques of available data, it may help in assessing past unknown years from available datasets. Factor models are based on assumed links between exogenic factors (such as population size, income level and gross domestic product "GDP") and waste generation [24]. Input–output analysis evaluates material flows routes from sources to final destination [30]. In this paper, the main objective of this study is to assess the potential amount of WEEE generated in Jordan during the period from 2000–2050 using, for the first time, a model that combines the population balance model (PBM), logistic growth model (LGM) and Weibull distribution.

### 1.1. Jordan WEEE Status

Jordan sits in the heart of the World at the crossroads of the continents of Asia, Africa and Europe. It is classified by the World Bank as an "upper-middle income" country. The economy, which has a GDP of USD 45.74 billion (as of 2021), grew at an average rate of 8% per annum between 2004 and 2008, and around 2.2% in 2021 with current GDP of USD 4103 per capita [31]. It is one of the smallest economies in the region, and the country's populace suffers from relatively high rates of unemployment and poverty [32]. Jordan imports most of its needs of EEE, with a small portion of products being manufactured locally. EEE Put on Market (EEE POM) increases steadily over time. It is expected that revenue in the household appliances segment is projected to reach USD 382 M in 2023 with an annual growth rate of 13% reaching USD 624 M in 2027 [33]. The average revenue per capita is projected around USD 200/capita in 2023. China dominates the countries exporting to Jordan by exceeding quarter 2023 revenues.

WEEE generation in the Arab States is relatively small, about 5% of 2019 figures according to Forti et al. [2]; however, in their regional report focusing on the Arab States, Iattoni et al. [7] reported that WEEE generation increased by more than 60% from 2010 to 2019, reaching 2.8 Mt (6.6 kg/inh) with growth rate greater than the global level. Saudi Arabia was the largest WEEE generator of 595 kiloton (kt) or 13.2 kg/inh, while Jordan is somewhere in the middle with about 55 kt, or 4.2 kg/inh, and Comoros is the lowest (0.6 kt, or 0.7 kg/inh). A total of 2.2 kt (0.01 kg/inh) of WEEE was collected for environmentally sound treatment in 2019 across only four Arab States, representing a collection rate of 0.1% of all WEEE generated in the region. Jordan had the highest WEEE collection rate of about 2.6%, or 0.1 kg/inh.

The realization of the WEEE problem in Jordan started in the late 2000s. It was firstly addressed, in academia, by some of the authors in the form of scientific projects to investigate the WEEE in Jordan. In parallel, several initiatives led mainly by voluntary agencies, NGOs and UN agencies, among others with the aim of raising awareness, organizing collection events and managing WEEE. The management of WEEE in Jordan is scattered and primitive. The obsolete EEEs are stored by households, offered for secondhand use, sold to collectors or discarded with municipal waste [34]. Collecting, dismantling and primitive treatment are mostly performed by informal sector in unregulated conditions that harm humans and the environment [34–36].

Jordan's economic transformation remains conditional on finding opportunities to widen the economy's outward orientation and to apply reforms needed to promote private sector-led growth and job creation [37]. The World Bank group has been a key partner on Jordan's reform agenda since 2018, and the launch of the Government of Jordan (GOJ) reform matrix, followed by economic priorities program (EPP) 2021–2023, and the launch of new Vision for Economic Modernization. These transformations and reforms are prioritizing key business environment reforms and activating Public Private Partnerships (PPPs) and financing in key sectors for investment and job creation, as well as promoting sustainable development agenda, green economy and environmental conservation.

In summary, this political and environmental background reveals that Jordan is moving in its early stages toward WEEE management. Thus, practical actions are still required to ensure that sustainable development goals of WEEE management are satisfied.

Behavior of consumers, EEE availability and prices are determining factors in the WEEE dilemma. Apparently, Jordan's market is open, and various EEE brands are available with a range of prices that makes them affordable for most households. In fact, studies addressing the WEEE problem in Jordan are scarce. In an early study, Fraige et al. assessed the awareness, behavior and willingness of Jordanian households regarding WEEE [34]. The findings of their study revealed that most Jordanians preferred to buy new appliances. They provided details about household consumption and disposal behavior once their EEE reached its EoL. They measured the average lifetime of EEE in the Jordanian community and roughly gave WEEE generation prediction.

Also, Fraige et al. summarized the WEEE problem in the MENA region, focusing on the environmental effects of WEEE informal recycling [35,36]. In the field of treating WEEE, Fraige et al. used mechanical vibration to separate shredded WEEE [36]. In other work, Alsheyab numerically determined the potential recovery of precious metals from high grade WEEE using mass flow analysis [38]. Ikhlayel examined the advantages and disadvantages of different WEEE generation methods [39]. He modified a method to estimate WEEE generation for developing countries based on the "consumption and use" method. Hamdan and Saidan evaluate the status of WEEE in Jordan by national statistics survey of about 16 k households, where primary data on WEEE generation and disposal methods were gathered, assessed and quantified [40]. They concluded that, in 2018, about 9 M EEE had turned into WEEE ($\approx$13 kt) in Jordan. This is equivalent to about 1.3 kg/inh, which is quite low compared to others' estimations [7,34,39].

Ikhlayel employed Life Cycle Assessment (LCA) to evaluate the environmental impacts and benefits of five different WEEE management systems, using Jordan as a case study [41]. He extended his studies to introduce integrated WEEE management, which theoretically combined the management of both municipal solid waste and WEEE for developing countries [42]. He claimed that this approach can improve the WEEE handling system in developing countries by tackling region-specific issues.

Jordan rules on WEEE aim to promote sustainable production, use and consumption of EEE. Its efforts to handle the WEEE problem were succeeded by approving WEEE management instruction in 2021. Its main aim is to protect the environment by utilizing the environmentally sound management system of WEEE. The instruction covers most EEE and its accessories, components and sub-parts. Any generated waste of this category is required to be disposed of in designated sites, and it cannot be discarded with municipal waste. While the import of WEEE is prohibited in this instruction, any WEEE export should be pre-approved by the Ministry of Environment [23].

To improve WEEE collection, treatment and recycling, eight private companies were established in Jordan, according to MoEnv officials [43]. By law, these companies are mandated to report the amounts and types of waste generated/collected/treated and exported to the MoEnv. They are also required to minimize the amount of waste generated from their activities by following the best available environmental practices. MoEnv officials reported that 27 t of WEEE are collected, sorted and, in some cases, exported by these companies every day [43]. This is close to 10 kt per annum. Improving WEEE collection, treatment and recycling can increase resource efficiency, improve sustainable production, use and consumption, and promote the circular economy in the country. This commitment towards better environment is reflected in the high environmental performance index that Jordan achieved compared with other Arab countries [44].

### 1.2. Motivation of the Study

Jordan, like some developing and underdeveloped countries, suffers from several enviro-economic, and infrastructural obstacles. These include lack of reliable WEEE inventory data, absence of regulations to define product responsibility, need for informational and technological systems, lack of financial support and lack of infrastructure, among others. For better WEEE management, these limitations should be overcome. The cornerstone in recognizing the size of WEEE and planning its subsequent management phases is

estimating current and predicting futuristic generation rates within data scarcity conditions. One of the reliable options to overcome this problem is utilizing modeling techniques to simulate the WEEE generation system. Hence, the particular importance of this paper lies in setting a firm foundation for straight and reliable techniques for WEEE generation rates suitable for countries like Jordan. For this purpose, modeling techniques were devoted to simulating temporal WEEE generation in Jordan under data constraints during the period from 2000–2050, using for the first time, a model that combines the population balance, logistic growth and Weibull distribution models.

As discussed by Kim et al., the pros of PBM are that it is originated from mass balance principles (shipment volume as inflow; waste volume as outflow; and ownership as stock) [45]. That means, it is less likely that WEEE is over- or under-estimated. Also, because it is a time-series material flow analysis model, it can estimate past and future WEEE generation. Nevertheless, the cons of this model are the difficulties when estimating products of fast growth phase or decline stage on the market, as the parameters of lifespan distribution for those products might change. For the latter case, transition lifespans of EEE are recommended (see for example, [46]).

## 2. Methods & Analysis

In this paper, three modeling techniques were combined in order to estimate the historical quantities of WEEE and to predict its future trends. It uses: (1) PBM to investigate the inflow and outflow of EEE quantities; (2) LGM to simulate the share of EEE in society and diffusion rates with time; and (3) Weibull distribution to estimate EEE lifetime distribution based on a questionnaire survey of Fraige et al. [34].

Other techniques, such as nonlinear least square fitting using generalized reduced gradient (GRG) solving method [47], and multi parameters regression analysis are employed as well. The following section provides more details about data sources, scope of the study, models and data processing.

### 2.1. Data Sources

Historical data about population and EEE are obtained from the Jordan department of statistics (JDOS) website [48]. This includes population and household size, percentage share of EEE possessed by households and any other statistics relevant to WEEE. The expected growth in population is extracted from 2022 World Population Prospects (WPP) up to 2050 using median growth prospect [49]. The import, export and production data were extracted from the harmonized commodity description and coding system of each selected EEE from the online JDOS [48] and UN Comtrade Databases [50]. The study used Fraige et al. [34] survey results for the average lifetime of EEE as summarized in Table 1. It also sorts studied EEE according to UNU-keys and equivalent EU-6 classification [2].

**Table 1.** Average lifetime [34] and LGM parameters of studied EEE.

| EEE | UNU-Keys (EEE Category under EU-6) | Average Lifetime (Years) | a | b | c | $R^2$ |
|---|---|---|---|---|---|---|
| TV | 0308, 0309 (Screens and monitors) | 10.5 | 0.52 | 0.11 | 1978 | 0.986 |
| R | 0108 (Temperature exchange equipment) | 11.8 | 0.34 | 0.1 | 1985 | 0.992 |
| W | 0104 (Large equipment) | 9.3 | 0.3 | 0.11 | 1988 | 0.99 |
| AC | 0111 (Temperature exchange equipment) | 8.3 | 80.87 | 0.17 | 1995 | 0.983 |

### 2.2. Scope of the Study

The selected EEEs in this study cover the main appliances such as: washing machines to represent large home appliances, TV for monitors and screens, refrigerators and air conditioners for temperature exchange equipment. During economic growth in developing countries, the diffusion of EEE has been generally observed in the order of TVs, Rs, Ws and ACs [51]. These appliances were chosen because they represent the expected highest percentage of Jordan's WEEE stream [34]. Also, they were selected by several researchers to approximate national WEEE generation rates, such as Kosai et al. [46]. In Japan, for example, successful recycling plants for these specified appliances are in action with high resource recycling rates of more than 94% [52,53]. Ideally, it is recognized that WEEE generation should include all EEE. However, the selected appliances represent the highest percentage by weight, with high rate of possession between households. Also, these appliances are bulky and need space for storage; hence, high collection rates are expected once they reach their EoL. While other items suffer from relatively low collection rates in comparison with the investigated EEE. Mobile phones, for instance, are easy to hoard in cabinets, drawers and homes [54]. In addition, people tend to keep other gadgets, such as personal computer and laptops, for data security reasons. Indeed, it is a limitation of the study, however, it is presumed that its effect will be minimal on WEEE estimation given data scarcity discussed earlier.

### 2.3. Population Balance Model

PBM is a widely used tool in science and engineering applications, including pharmaceutical manufacture and system growth [55]. Tasaki et al. [56,57], and more recently Kosai et al. [46] used PBM to predict WEEE generation. This model is successfully applied to estimate WEEE in South Korea [45] and Vietnam [46]. In this model, it is assumed that the change in appliance numbers possessed between two successive years ($N_T$ and $N_{T-1}$) is balanced by the resultant of total shipment ($S_T$) minus the total number discarded ($D_T$) in year $T$. It is expressed, in the context of WEEE, by the following equation [56,57]:

$$N_T - N_{T-1} = S_T - D_T \tag{1}$$

#### 2.3.1. Household Number and Size

The number of EEE possessions is usually obtained from inventory statistics. However, in cases where there is data scarcity on EEE inventory, sales and disposal, especially in developing countries like Jordan, it is inevitable to rely on assumptions that may facilitate the estimations. As the number of EEE possession is a measure of how much an EEE is diffused in a country, then one may relate this to the number of households in the country and multiply by average EEE possession ratio [39,58]. In this regard, historical, as well as predicted, future figures are needed to reflect temporal changes of population and household size (HS). The availability of HS is limited to specific years at which statistical surveys were conducted. It is observed from the available data that HS reduces with time despite population growth in Jordan as illustrated in Figure 1. So, to estimate the household number (H), it is required to predict the household size for the study period. Here, regression analysis is used to obtain HS with time given the effect of socioeconomic parameters, such as population size and GDP.

Now, the number of households ($H$) can be determined using population ($P$) and *HS* data at the corresponding year as given below:

$$H = \frac{P}{HS} \tag{2}$$

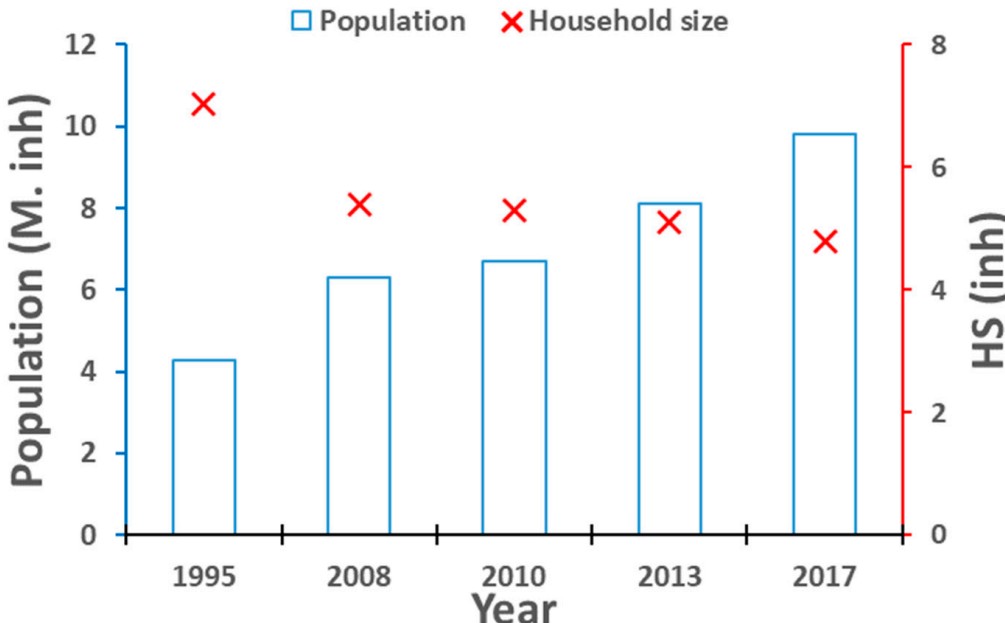

**Figure 1.** Relationship between population and household size in Jordan. Data were extracted from several annual reports and statistics obtained from [48]. However, the graph is published for the first time.

### 2.3.2. $N_T$ Estimation

In order to predict $N_T$, the average household possession of EEE can be utilized to approximate the number of appliances. JDOS measured the share of household's appliances during its national family surveys. However, these are limited in number, when surveys are conducted and more details about the diffusion is needed. Share percentage of appliances possessed per household ($n$) is demonstrated here using LGM. This model is usually used for modeling data where drastic change in rate of growth is observed. The trend of the EEE demand could be approximated using this model [46]. Predicted diffusion rates of EEE using LGM usually follow S-shape function. It has an increasing growth rate with time at the beginning, followed by uniform rate, then decreasing growth at a later stage as it gets closer to a maximum.

The share percentage of appliances possessed per household ($n$) was estimated by the following logistic growth model [59]:

$$n = \frac{n_m}{1 + a\, e^{-b(T-c)}} \qquad (3)$$

where $n_m$ is the maximum equipment share percentage per household (in this work, it is assumed 100); $a$, $b$, and $c$ are constants; and $T$ is time variable. The constants $a$, $b$ and $c$ are resolved to approximate the available historical data [48]. Nonlinear least square fitting using GRG solving method is employed to optimize these parameters [47].

Number in possession of each appliance type at specific year ($N_T$) can be estimated now by multiplying the share percentage of appliances per household ($n$) by the number of households ($H$) as given below:

$$N_T = nH \qquad (4)$$

### 2.3.3. $S_T$ and $D_T$ Estimation

EEE shipment (EEE POM) figures are obtained from past data collected from previous statistics and literature as indicated in data sources. Future shipment figures are predicted from the PBM in a systematic manner as it will be explained below.

The disposal of EEE could be correlated with EEE purchased and its average lifetime to disposal. Lifetime data of each appliance can be extracted from different sources, such as ideal expected lifespan of EEE, employing data about EEE lifetime from communities having similar features, or conducting surveys to measure lifetime in the same society. In this work, average lifetime of appliances [34] with the introduction of probabilistic distribution is used. Normal, Gaussian lognormal and Weibull distributions are examples of different distributions that can be used [60]. Wang et al. recommended the use of the Weibull distribution model to simulate the EEE lifetime [30]. This distribution is utilized in this work as described below [59]:

$$d_t = \frac{\beta}{\alpha}\left(\frac{t}{\alpha}\right)^{\beta-1} e^{-\left(\frac{t}{\alpha}\right)^{\beta}} \tag{5}$$

where $d_t$ is the disposal ratio of EEE at certain age $t$ and $\alpha$ and $\beta$ are the Weibull distribution parameters. The scale parameter ($\alpha$) is the lifetime of each EEE. While shape parameter ($\beta$) is assumed 3.44 (i.e., mean = median for normal distribution [61]. The discarded number of EEE at specific year $T$ is estimated from the discarded EEE from the year that the appliance is put on market ($T_p$) to year ($T-T_p$) assuming ($d_t = 0$) when ($T = T_p$). This will facilitate computation especially for future predictions of EEE POM based on EEE demand and PBM. The discarded number of EEE may be written as [59]:

$$D_T = \sum_{T_p < T} S_{T_p} \cdot d_{(T-T_p)} \tag{6}$$

### 2.4. Data Processing Procedure

The available historical data of EEE POM, starting from 2000 as a base year, is used to calculate the discarded quantities based on disposal ratio using Equations (5) and (6) up to year 2021. The future demand of EEE for two successive years is estimated from Equation (4). Then, given that demand and discarded quantities are set, expected future year $S_T$ (EEE POM) is calculated using Equation (1) systematically year by year.

## 3. Results & Discussion

### 3.1. Predicting Household Number and Size

Household size data is an important parameter in WEEE prediction using this methodology. There are many socioeconomic and temporal factors that may be responsible for HS variation, such as population size and growth, income, social factors, time, among others. Here, HS is correlated with population size, GDP and time as explained earlier. The results obtained for HS variation with time using regression analysis are shown in Figure 2. A 10% HS reduction rate per year between 1995 and 2017 is observed. The predicted HS for the current year (2023) is 4.41. The decline in HS is expected to continue to reach 3.54 and 1.14 inhabitant in 2030 and 2050, respectively. This is mainly due to changing lifestyle, adopting family planning programs, increasing living costs and work conditions alteration.

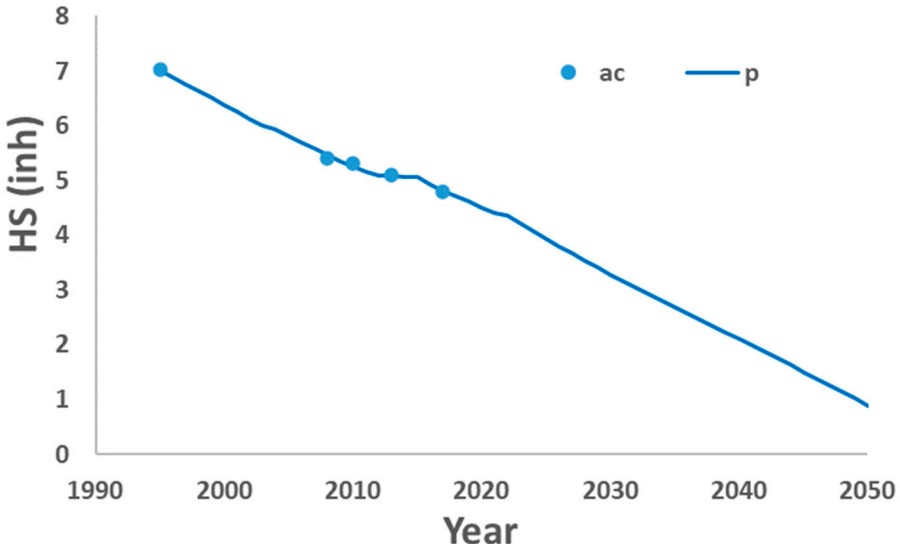

**Figure 2.** Comparison between actual (ac) and predicted (p) household size in Jordan. Data points (ac) were extracted from several annual reports and statistics obtained from [48].

*3.2. LGM Results*

Share percentage of appliances possessed per household ($n$) is approximated using LGM. Nonlinear least square fitting is used to optimize LGM parameters to approximate the available statistical data. Table 1 summarizes the optimized parameters for each appliance. Model prediction and actual data variation of appliance ratio possessed per household are compared as illustrated in Figure 3. The model predicts well the diffusion of appliances given the coefficient of determination $R^2 > 0.98$ for all cases.

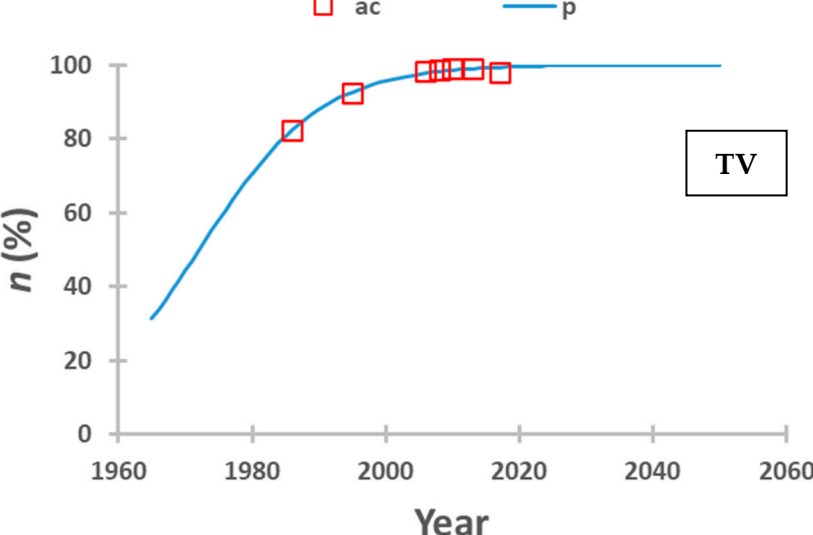

**Figure 3.** *Cont*.

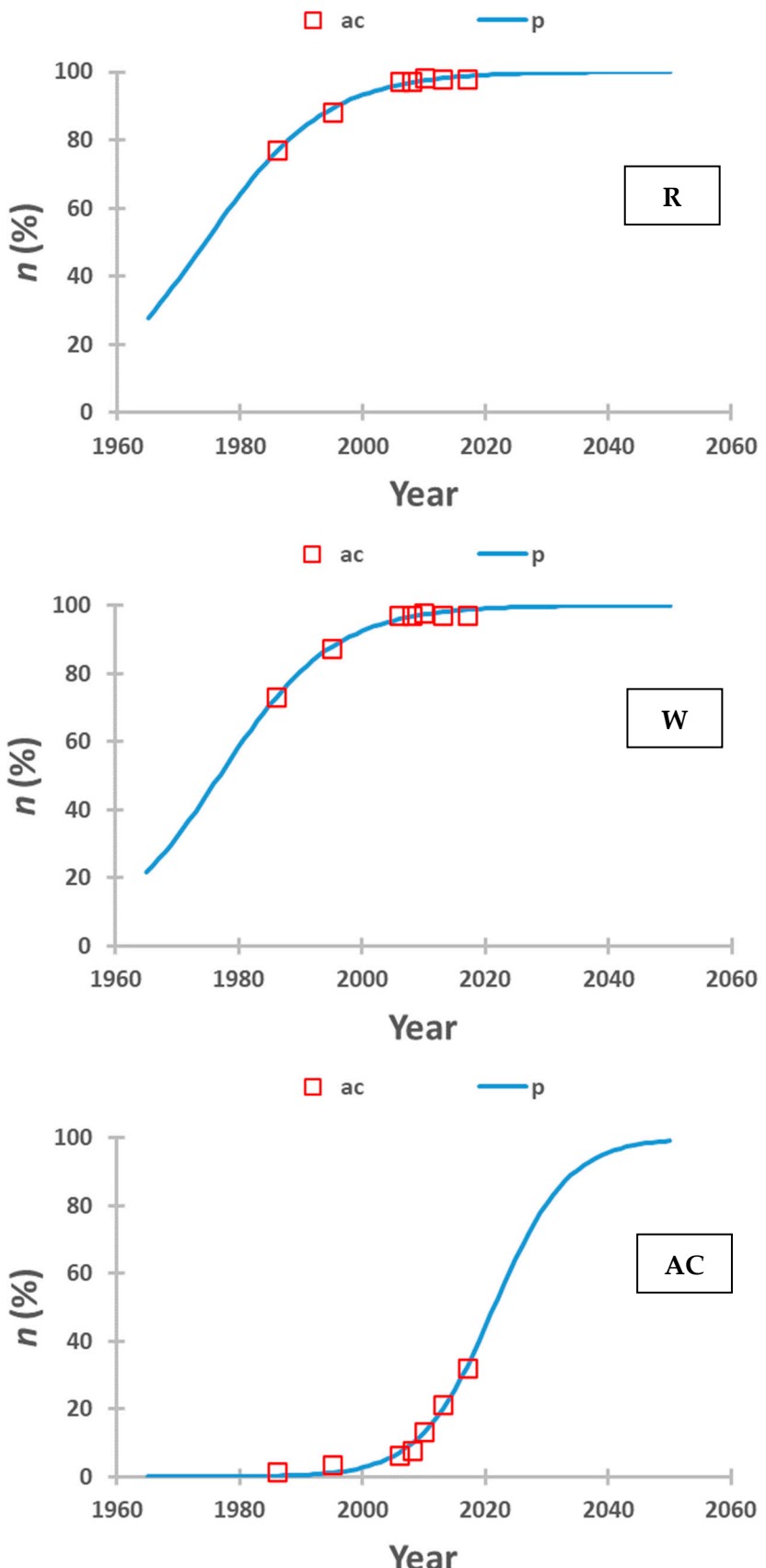

**Figure 3.** Fitting EEE possession ratio (Actual (ac) and predicted (p)) using LGM. Data points (ac) were extracted from several annual reports and statistics obtained from [48].

To facilitate the comparison of diffusion of various appliances, model prediction for the four devices is illustrated in Figure 4. While TV, R and W are reaching their saturation level (i.e., almost all households have these appliances), AC is in the stage of high diffusion rates, which indicates that the AC market is unsaturated. Multiple factors could play into this, such as technological innovation in AC industries which leads to more energy efficient equipment. Also, global warming and climate change in Jordan drives toward more demand for air conditioning units to combat the heat waves, where this commodity is no longer considered luxury. Other socioeconomic factors, like increasing GDP per capita, changing consuming behavior and lifestyle patterns, play a role as well.

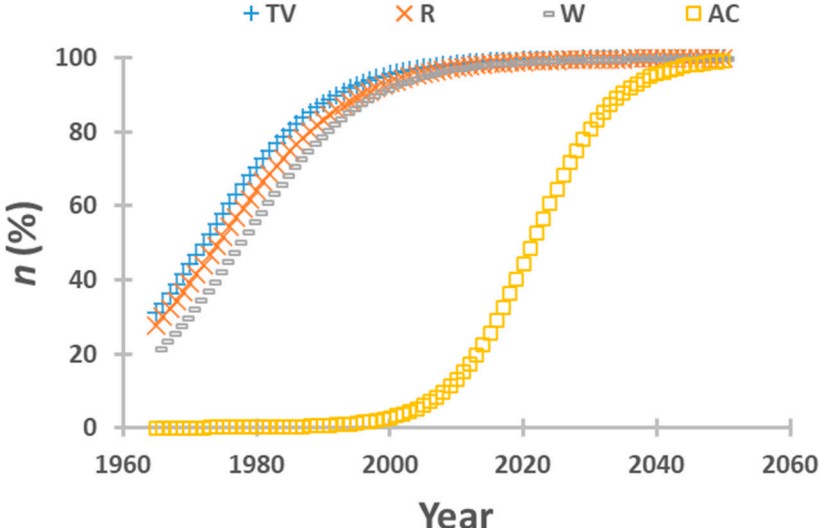

**Figure 4.** Predicted temporal possession ratio for the studied EEE.

### 3.3. Disposal Ratio of EEE

Disposal ratio is determined using average EEE lifetime and Weibull probabilistic distribution function. The predicted variation of disposal ratio of each appliance with EEE age is shown in Figure 5. Disposal of appliance is relatively small for new EEE; it increases as EEE gets older and reaches its maximum around EEE mean lifetime before it diminishes with time. Due to its shortest lifespan, air conditioners become obsolete faster than other EEE, followed by washing machines, televisions and refrigerators.

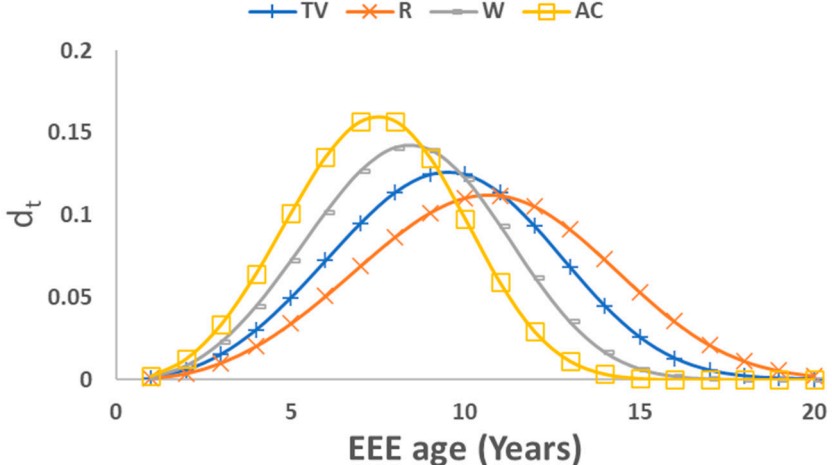

**Figure 5.** Disposal ratio of EEE obtained using Weibull Distribution.

### 3.4. Estimation of WEEE Generated in Jordan

The generated WEEE from the studied appliances is estimated in Jordan up to 2050. Generally, the waste increases with time as illustrated in Figure 6. This increase apparently is characterized by two growth rates: the first is uniform increase of about 2.4 kt per year up to 2030, while the other is higher with a mean rate of about 5.8 kt per year afterwards. It is expected that the total disposal of appliances reaches about 1.6 million units in 2022. This is equivalent to around 53 kt. The number of discarded EEE is predicted to double its 2022 figures by year 2044, recording more than 117 kt (an increase of more than 120% by weight).

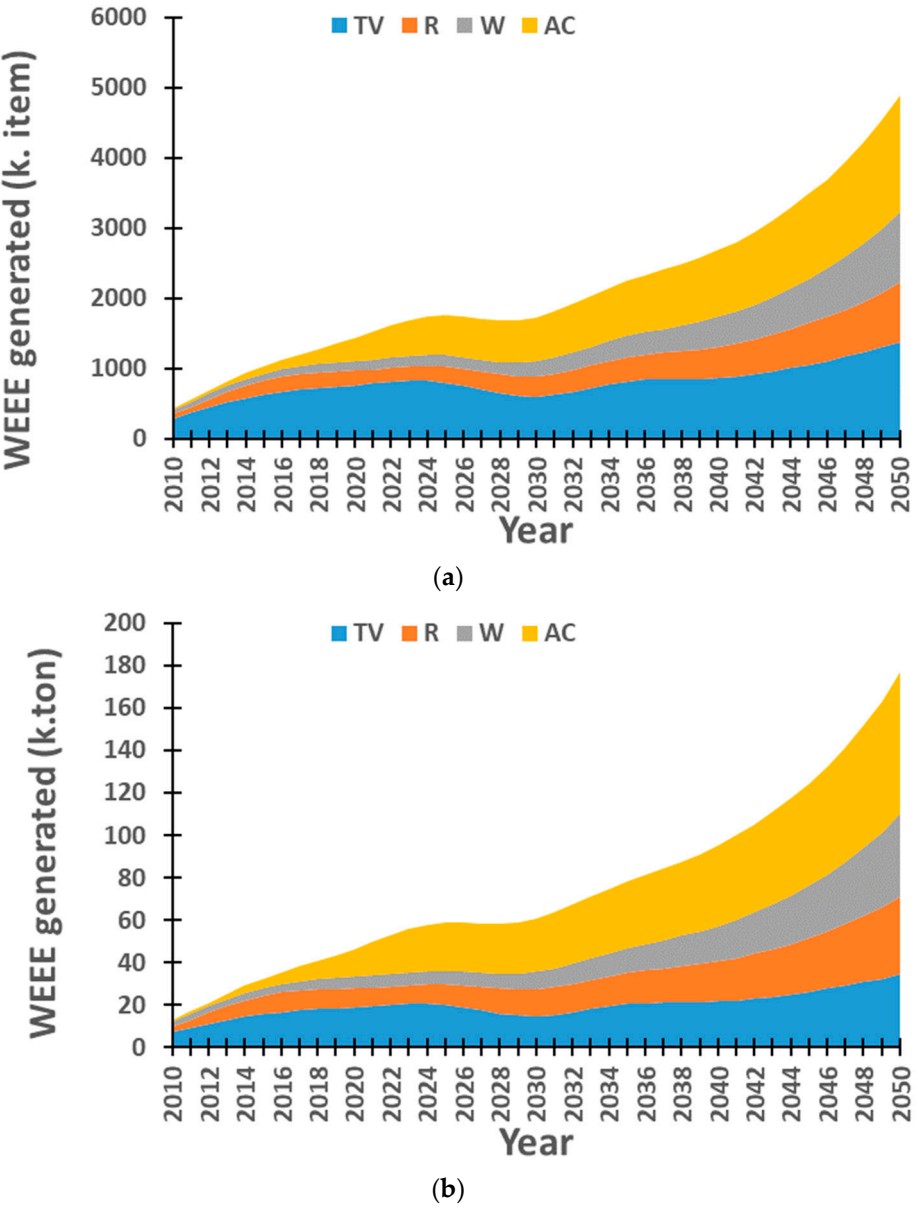

(**a**)

(**b**)

**Figure 6.** Generated WEEE quantities in Jordan by number (**a**) and by weight (**b**).

The demand for EEE in the period of 2010–2012 consisted of 33% TV, 18% R, 19% W, 30% AC by weight. The strong demand for AC units is apparent as explained before, coupled with high demand for replacing outdated TVs with modern units due to relatively reduced prices and advanced technology. The composition of WEEE generated in the same period was: 54% TV, 23% R, 16%W, 7% AC. Ten years later WEEE generation recorded 3.9 kg/inh with changes in WEEE composition (38% TV, 16% R, 11% W, 35% AC). Discarded

TV, R and W percentages were reduced in favor of the increasing share of AC units as it would be expected from the high demand and shortest lifespan for AC units.

Another important indicator is the quantity of EEE POM and its effect on WEEE generation in Jordan. Figure 7 shows the relationship between EEE POM and WEEE generated for the period 2010–2022. It is observed that during this period, EEE POM per inhabitant passes through two distinct phases: increasing stage from 2012 to 2017, followed by a decreasing stage to 2021. In the first stage, this increase has been driven mainly by the flood of about 1.4 million Syrian refugees fleeing their country due to the Syrian conflict started in 2011 [62]. This caused an increased demand for EEE in Jordan from 3 to about 8 kg/inh. On the contrary, when the COVID-19 pandemic spread in 2019, the EEE demand of investigated appliances was reduced to about 2.6 kg/inh. This is due to the consequences of the pandemic on Jordanian households causing some socioeconomic effects such as reducing income due to losing jobs or reducing wages and customers' reaction represented with tendency toward saving rather than spending money for buying new appliances, especially in the uncertain circumstances they passed through. However, signs of market pack up are observed in 2022 where EEE POM reached about 5 kg/inh as illustrated in Figure 7. The dynamics of market is captured by the PBM (in terms of delayed WEEE) as shown in Figure 6.

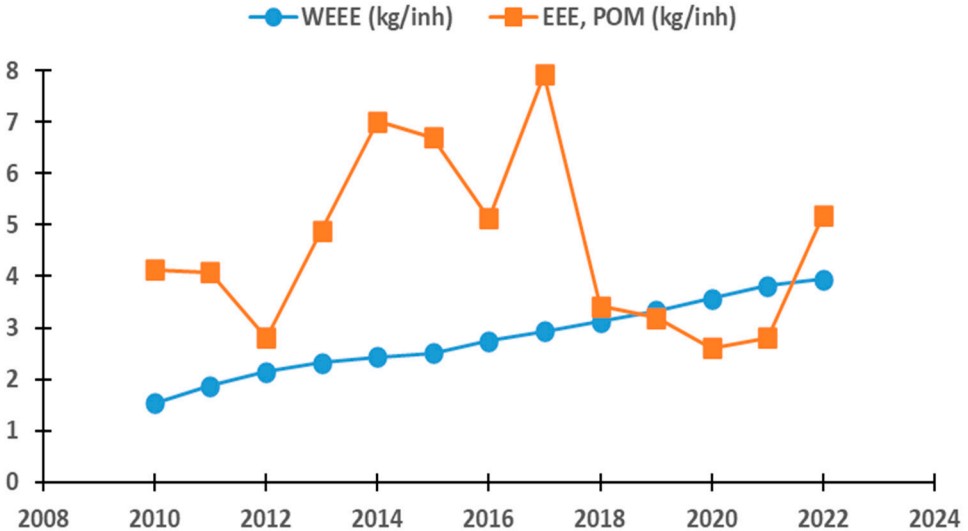

**Figure 7.** Variation of EEE POM and generated WEEE per inhabitant with time. Data points were extracted from several annual reports and statistics obtained from [48,50].

Jordan EEE POM and WEEE performance are compared with figures from other countries in the regional and international levels as illustrated in Table 2. Arab States in wealthy oil-producing countries, such as Saudi Arabia and UAE, have the highest EEE POM with more than 20 kg/inh, while African states like Djibouti and Comoros have the lowest of less than 2 kg/inh. Jordan is in the middle and below EEE POM average of the Arab states.

Model results obtained here are verified with results from other sources. Predicted WEEE per capita generated in this work is compared well with the average predictions of other sources. This indicates that this model and methodology are suitable for predicting WEEE figures for developing countries, where data are scarce.

**Table 2.** EEE POM and WEEE statistics compared with this study.

| Country | Year | EEE, POM (kg/inh) | WEEE (kg/inh) | Source |
|---|---|---|---|---|
| Bahrain | 2019 | 22 | 15.9 | |
| Comoros | 2019 | 0.8 | 0.7 | [7] |
| Djibouti | 2019 | 1.9 | 1 | |
| Egypt | 2019 | 10.8 | 5.9 | |
| | 2010 | 4.1 | 1.5 | [34] |
| | 2013 | NA | 3–4.8 | [39] |
| Jordan | 2018 | 7.6 | 5.3 | [7] |
| | 2018 | 7.6 | 1.3 | [40] |
| | 2022 | 5.2 | 4 | This work |
| Lebanon | 2019 | 10.3 | 8.2 | |
| Morocco | 2019 | 5.9 | 4.6 | |
| Saudi Arabia | 2019 | 22.4 | 17.6 | |
| Sudan | 2019 | 2 | 2.1 | [7] |
| Tunisia | 2019 | 7.6 | 6.4 | |
| United Arab Emirates | 2019 | 24 | 15 | |
| Arab States Average | | 9.53 | 6.59 | |
| Asia Average | 2019 | | 5.6 | |
| Africa Average | 2019 | | 2.5 | [2] |
| Europe Average | 2019 | | 16.2 | |
| Americas Average | 2019 | | 13.3 | |

## 4. Conclusions

In this paper, a model based on population balance, logistic growth and Weibull distribution is employed to assess the WEEE in Jordan. It covers the main appliances such as washing machines to represent large home appliances, TV for monitors and screens, refrigerators and air conditioners for temperature exchange equipment. Past data about EEE POM and WEEE are analyzed, and future trends are predicted. Appliance possession number is approximated using logistics growth model up to 2050, while future demand and disposal predictions are estimated using population balance model and Weibull distribution. It is expected that the total disposal of appliances reaches about 1.6 million units (53 kt) in 2022 and doubles this figure by 2044 with changing waste composition. This suggests rapid increase of WEEE in the near future and needs urgent reaction to handle this waste.

Jordan EEE POM and WEEE performance are relatively in the middle rank with figures slightly below average in comparison with Arab countries. The results of the model are verified by comparing with the predictions of other studies found in literature, indicating the suitability of this model for predicting WEEE in developing countries where data is scarce.

Tackling the WEEE issue in Jordan is still in the premature level, and areas of research and development are open wide. It is recommended to keep reliable WEEE inventory data, establishing efficient management systems, timely collection of WEEE, and applying 3R initiative (reduce, reuse, recycle) in sustainable circular supply chain design of EEE is of inevitable importance.

**Author Contributions:** Conceptualization, F.Y.F. and L.A.A.-K.; Data curation, F.Y.F., L.A.A.-K. and M.A.A.-S.; Formal analysis, F.Y.F.; Investigation, F.Y.F. and M.A.A.-S.; Methodology, F.Y.F. and L.A.A.-K.; Resources, L.A.A.-K. and M.A.A.-S.; Software, F.Y.F.; Validation, F.Y.F.; Writing—original draft, F.Y.F. and L.A.A.-K.; Writing—review & editing, F.Y.F., L.A.A.-K. and M.A.A.-S. All authors have read and agreed to the published version of the manuscript.

**Funding:** This research received no external funding.

**Institutional Review Board Statement:** Not applicable.

**Informed Consent Statement:** Not applicable.

**Data Availability Statement:** Data used in this study is available as described in Section 2.1 Data Sources. For further information, reader is courteously encouraged to contact the corresponding author.

**Acknowledgments:** The authors would like to thank the anonymous reviewers for their valuable inputs.

**Conflicts of Interest:** The authors declare no conflict of interest.

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
