# Peer review of "Predicting WEEE Generation Rates in Jordan Using Population Balance Model"

_sustainability, doi:10.3390/su15032845_

Round 1

Reviewer 1 Report

Abstract.

What do you mean by “POM”?

Line 29“Radha, 2002; DIT, 2003”.

These references appear too dated.

Lines 30-33: “This happens to satisfy the cumulative demand and use of information and communication technologies (ICT), renewable and green energy products, electric vehicles, and smart cities, which reached an alarming rate of about 2.5 million tons (Mt) per annum (Forti et al., 2020).”

Your sentence is unclear. In the manuscript you cited as a reference (i.e. Forti et al., 2020), the authors wrote as follows: “On average, the total weight (excluding photovoltaic panels) of global EEE consumption increases annually by 2.5 million metric tons (Mt).” Their sentence is much clearer than yours.

Section 1. Introduction.

There are two main weaknesses in the introduction:

- You need to improve the whole grammar;

- You need to mention more research that has been done on this topic, in particular, at the international level.

Lines 143-144“three modelling techniques were combined in order to estimate the historical quantities of WEEE and predict its future trends.”

It is necessary to discuss the reliability of the chosen techniques. Have previous authors used them?

Section 2. Methods & Analysis.

The section is well done. However, the structure is unconventional. Indeed, research articles usually have a Methods section and then a Results section. Thus, please put the results you obtained in this section in the following section (that you have correctly named “Results”). For example, Figure 3 and Figure 4 represent the results that you obtained in your work, right?

Lines 281-282“Figure 6Error! Reference source not found”.

Additional consideration/limitations.

Why did you not include mobile phones (including smartphones) in your research? According to some studies, they may represent about 10% of global E-waste. Thus, you should discuss this “limitation” of your research in detail, giving adequate explanations.

Author Response

We would like to thank you for your positive and insightful comments on the manuscript. Please, kindly find our response to the issues raised in the review in the attached file.

Reviewer 2 Report

I appreciate the contribution of the Authors and the development of the model, however I lack a better background image - availability, prices and accessibility for the population; trends on E&E market; consumer behavior in the field of E&E - the possibility of easy repair, the method of collecting electrical and bulky waste; the impact of legislative changes on the behavior (design) of devices and on the consumers, ctc. The model reflects the market situation as well as it takes into account additional market factors.

It is worth pointing out potential areas of using these analyses, their strengths and weaknesses. Apart from the complicated mathematical modelling, I am not convinced of the usefulness of this study. Please describe the motivation for taking up the topic, the political and environmental background of the problem and the effects of this article.

Author Response

(The authors gave the same response as above.)

Round 2

Reviewer 1 Report

Well done